# Easy Expression and Purification of Fluorescent N-Terminal BCL11B CCHC Zinc Finger Domain

**DOI:** 10.3390/molecules26247576

**Published:** 2021-12-14

**Authors:** Anne Susemihl, Felix Nagel, Piotr Grabarczyk, Christian A. Schmidt, Mihaela Delcea

**Affiliations:** 1Department of Biophysical Chemistry, Institute of Biochemistry, University of Greifswald, 17489 Greifswald, Germany; anne.susemihl@uni-greifswald.de (A.S.); felix.nagel@uni-greifswald.de (F.N.); 2Department of Hematology and Oncology, Internal Medicine C, University of Greifswald, 17489 Greifswald, Germany; piotr.grabarczyk@med.uni-greifswald.de (P.G.); christian.schmidt@med.uni-greifswald.de (C.A.S.)

**Keywords:** BCL11B, FRET, homodimerization, protein expression and purification, zinc finger

## Abstract

Zinc finger proteins play pivotal roles in health and disease and exert critical functions in various cellular processes. A majority of zinc finger proteins bind DNA and act as transcription factors. B-cell lymphoma/leukemia 11B (BCL11B) represents one member of the large family of zinc finger proteins. The N-terminal domain of BCL11B was shown to be crucial for BCL11B to exert its proper function by homodimerization. Here, we describe an easy and fast preparation protocol to yield the fluorescently tagged protein of the recombinant N-terminal BCL11B zinc finger domain (BCL11B_42-94_) for in vitro studies. First, we expressed fluorescently tagged BCL11B_42-94_ in *E. coli* and described the subsequent purification utilizing immobilized metal ion affinity chromatography to achieve very high yields of a purified fusion protein of 200 mg/L culture. We proceeded with characterizing the atypical zinc finger domain using circular dichroism and size exclusion chromatography. Validation of the functional fluorescent pair CyPet-/EYFP-BCL11B_42-94_ was achieved with Förster resonance energy transfer. Our protocol can be utilized to study other zinc finger domains to expand the knowledge in this field.

## 1. Introduction

Proteins with domains that participate in zinc ion (Zn^2+^) binding are widely represented across eukaryotic genomes. About 10% of the encoded proteins within the human genome participate in such zinc binding, and a large part of these proteins are known as zinc finger proteins [1]. The most prominent zinc finger motif is the C2H2 coordination environment, where two cysteines and two histidines are involved in zinc binding [2]. These typical C2H2 zinc fingers are best known for interacting with nucleic acids in a sequence-specific manner and, thus, exerting critical functions in gene expression [3,4]. Based on these abilities, many zinc finger proteins are transcription factors (TFs). Due to their role in several cellular processes, zinc finger proteins play key roles in various diseases, as the recent data highlights. Since they function as both oncogenes and tumor suppressor genes [5,6], altered expressions of specific zinc finger proteins is found, for example, in various human cancers, Parkinson’s disease, and congenital heart disease [5,7,8]. 

A clinically relevant protein, especially in the context of hematologic diseases [7,8], is B-cell lymphoma/leukemia 11B (BCL11B). BCL11B is known as a Krüppel-like transcription factor and possesses six C2H2 zinc finger domains [9]. The zinc complexing domains two and three are responsible for DNA binding [10]. BCL11B appears to be relevant for the differentiation of various tissues and organs, as well as in T-cell development [11]. As shown in multiple mouse models, development of the central nervous system, skin, or teeth is critically perturbed upon *Bcl11b* deletion [12,13,14,15]. Such dysregulation in *BCL11B* expression is associated with multiple pathologies, like different cancer types, and neurodegenerative disorders [16,17]. Recently, an additional zinc finger domain located at the N-terminus of the BCL11B protein showing atypical CCHC zinc finger properties was described. This domain was revealed to be crucial for the dimerization of BCL11B and, thus, for its proper function [18].

Here, we established an easy expression and purification protocol for the fluorescently tagged N-terminal zinc finger domain of BCL11B as a model system. 

## 2. Results

### 2.1. Construct Design and Expression of Fluorescently tagged BCL11B_42-94_

In order to overexpress the N-terminal zinc finger domain of BCL11B from amino acids 42-94 (BCL11B_42-94_), a customized vector was designed. The vector contained an ampicillin resistance gene, an N-terminal hexahistidine tag for purification, fluorescent protein tags (CyPet and EYFP, respectively), a linker to render the protein construct flexible, and a TEV-cleaving site to obtain untagged BCL11B_42-94_ (Figure 1A). The amino acid sequence for CyPet-BCL11B_42-94_ is found in Appendix A. After TEV cleavage, the untagged domain contains three additional amino acids (Gly, Leu, and Glu) at its N-terminus. Introduction of the flexible peptide linker (GGGGS)_3_ is crucial for the subsequent FRET assays ensuring sufficient energy transfer. Additionally, the flexibility of the fluorescent protein tag limits interference with the BCL11B–BCL11B interaction. 

Fluorescent proteins derived from green fluorescent protein (GFP), like CyPet and EYFP, tend to dimerize at higher concentrations [19,20], which can be prevented by introducing a mutation at position 206 [21]. The A206K mutation was introduced by site-directed mutagenesis for EYFP (Appendix A), while it was already present in CyPet.

Overnight cultures of CyPet-BCL11B_42-94_ and mEYFP-BCL11B_42-94_, respectively, were used to inoculate the TB medium. Cultures were grown to an OD_600_ = 2. After induction, the cultures were grown at 37 °C and 16 °C, respectively (Figure 1B,C). Optimizing the expression temperature turned out to be crucial for protein solubility. Growth at 37 °C led to an insoluble protein that was found exclusively in the bacterial pellet (Figure 1B). Growth at 16 °C increased the solubility of the fluorescently tagged BCL11B_42-94_, indicated by the intense bands at 36 kDa (Figure 1C). Overexpression was carried out fast, as the band intensity did not visibly increase after 1 h of induction (Figure 1C). As expected, at 16 °C, the overexpression of BCL11B_42-94_ was slower, indicated by a lesser contrast but increased solubility of the overexpressed protein (lanes at 36 kDa) and other proteins of *E. coli*. 

### 2.2. Purification of Fluorescently Tagged BCL11B_42-94_

After expression, the pellets were resuspended and lysed using sonication. The clarified lysate was loaded onto connected HisTrap excel columns (Figure 2A and Appendix A). The wash steps, especially the wash using 5% elution buffer, were essential to elute the impurities before the protein of interest and, thus, to achieve a high purity of the fluorescently tagged BCL11B_42-94_ domain (Figure 2B and Appendix A). The purified BCL11B_42-94_ domain showed strong fluorescence (Figure 2C). The expression and purification procedures yielded 0.2 g purified, fluorescently tagged zinc finger per liter of bacterial culture, regardless of which fluorescent tag was used.

### 2.3. Preparation of Untagged BCL11B_42-94_ Zinc Finger Domain

The designed construct enabled the cleavage of the fluorescently tagged zinc finger to obtain the untagged BCL11B_42-94_ domain. The final cleavage product contained three additional amino acids (Gly, Leu, and Glu) at the N-terminus that were not present in the native BCL11B sequence. Purified fluorescently tagged BCL11B_42-94_ was digested using TEV protease at room temperature overnight. Removal of the cleaved fluorescent tag was achieved through purification by immobilized metal affinity chromatography (IMAC; Figure 3A and Appendix A). The fluorescent tag bound to the column through its His_6_-tag, whereas the BCL11B_42-94_ domain was eluted. Absorption at 230 nm was recorded to monitor untagged BCL11B_42-94_, as HEPES, present in the buffer system, exhibited high absorption at a wavelength of 214 nm, which is usually used to record the absorption of the peptide bonds. The BCL11B_42-94_ domain itself lacks aromatic amino acids and, thus, did not absorb at 280 nm. The absorption maximum of the CyPet protein was 434 nm and could be used to monitor the absorption of the fluorescent tag. The collected fractions were analyzed by tricine SDS-PAGE (Figure 3B and Appendix A). The fractions containing untagged BCL11B_42-94_ (red) showed high purity. The initial elution fractions (black) still contained minor impurities, whereas the column wash (blue) did not contain BCL11B_42-94_. In total, the cleavage of 8 mg tagged BCL11B_42-94_ yielded 1 mg untagged BCL11B_42-94_.

### 2.4. Characterization of BCL11B_42-94_ Zinc Finger Domain Using Circular Dichroism (CD) and Förster Resonance Energy Transfer (FRET)

To verify the correct folding of the expressed zinc finger domain, CD spectra of the untagged zinc finger domain (Figure 4A,C) and the fluorescently tagged BCL11B_42-94_ (Figure 4B) were recorded. The spectra of the untagged N-terminal BCL11B zinc finger domain showed the typical minima at 208 and 228 nm characteristic for α-helices [22,23,24], as well as a maximum at 195 nm. Our obtained spectra verified the typical ββα-fold in the presence of zinc. Figure 4B shows the large influence of the fluorescent tag to the BCL11B_42-94_ spectrum. CyPet-BCL11B_42-94_ and mEYFP-BCL11B_42-94_ (Appendix A) showed very similar spectra and indicated characteristic β-sheet structures [22,23,24]. The secondary structure content obtained after spectra deconvolution (Table 1) demonstrated the high similarity of CyPet-BCL11B_42-94_ and mEYFP-BCL11B_42-94_. Minor differences in the β-sheet contents were most likely the result of deconvolution artefacts.

To validate the successful expression and purification of the FRET pair CyPet/mEYFP, the ability of BCL11B_42-94_ to execute subunit exchange was monitored by fluorescence. It is known that BCL11B tends to homodimerize to execute its function [18]. The subunit exchange could be monitored by FRET (Figure 4D). Pure mEYFP-BCL11B_42-94_ (blue) only showed minimal fluorescence at 530 nm, with an excitation at 400 nm. CyPet-BCL11B_42-94_ exhibited strong fluorescence, especially at 480 nm, when excited at the same wavelength (red). The 1:1 mixture of CyPet-BCL11B_42-94_ and mEYFP-BCL11B_42-94_ (black) showed its maximum fluorescence at 530 nm. This indicated an energy transfer from CyPet to mEYFP. This was possible because of the close proximity due to oligomerization of the zinc finger domain. There was also decreasing fluorescence at 480 nm of the mixture compared to pure CyPet-BCL11B_42-94_, which was due to donor quenching. This approach verified the successful expression of the CyPet/mEYFP FRET pair and supported the expectation that BCL11B at least forms homodimers. 

Size exclusion chromatography (SEC) was utilized to verify the homodimerization observed by the FRET assay. Chromatograms of CyPet-BCL11B_42-94_ (Figure 4E) revealed high purity and homogeneity of the purified fusion protein shown by a single peak at 1.35 mL, which was also observed for mEYFP-BCL11B_42-94_ (Appendix A). The elution volume corresponded to a size of ~150 kDa (Appendix A and Appendix A), indicating dimerization and potentially the formation of higher-order oligomers, thereby verifying the results of the FRET assay. Two smaller peaks at 1.6 mL and 1.7 mL corresponded to monomeric CyPet-BCL11B_42-94_ and solely the CyPet tag. The use of protease inhibitors during lysis might reduce the amount of cleaved CyPet tag. Size exclusion chromatography of the untagged BCL11B_42-94_ domain after cleaving CyPet-BCL11B_42-94_ (Figure 4F) and mEYFP-BCL11B_42-94_ (Appendix A) also indicated the formation of tetramers.

## 3. Discussion

For a better understanding of protein interactions, structural properties, and ligand binding, the production of soluble and functional recombinant proteins is crucial. While, for small proteins or peptides, both chemical synthesis and *E. coli* expression systems provide a unique set of advantages and disadvantages [27,28,29,30,31,32,33], we decided on a bacterial expression system for the following reasons: The transfer of the designed constructs into biological systems is easier and the data more comparable. Additionally, applications requiring a coating of the proteins, i.e., ELISA, need larger tags so that binding sites and epitopes are not hidden by the coating process. Several fusion tags might improve the solubility during expression but can also lead to various problems. For instance, GST tags tend to dimerize, which would interfere with the oligomerization analysis of BCL11B_42-94_. GFP-derived fluorescent protein tags enable the use of technologies like DNA microarrays without extra labeling steps. Therefore, they offer greater flexibility compared to other tags.

Zinc finger domains are involved in a plethora of biological processes, participate in many cellular pathways, and exert crucial functions therein. Perturbed zinc finger protein expression, e.g., due to mutations or translocations, is associated with several diseases [34]. In hematology, BCL11B mutations and altered expressions are correlated with T-cell lymphoblastic leukemia (T-ALL) [16,35]. Further insights into zinc finger domains will allow a better understanding of disease developments and could lead to better treatment options. The inhibition of transcription factors through the blocking of protein–protein interactions (PPIs), including oligomerization, was shown to be successful for several TFs—for instance, for the interaction of p53/HDM2 [36]. Taking advantage of the ability of zinc fingers to bind specific DNA sequences, designed zinc finger domains represent a potential target structure for gene-based therapeutic approaches in various genetic diseases [37,38,39]. Especially, artificial zinc finger nucleases find use in biotechnology and medical research. These synthetic restriction enzymes are able to target specific DNA sequences and subsequently alter the genomes of higher organisms [40].

Circular dichroism spectroscopy was used to verify the proper zinc finger folding of BCL11B_42-94_. The secondary structure of CCHC zinc finger proteins contains the typical ββα-fold, where a turn connects two short β-strands and is followed by an α-helix [22]. Our deconvolution data suggested such zinc finger formation in the presence of zinc. Removing Zn^2+^ from the zinc finger resulted in a change in the CD spectra, with BCL11B_42-94_ not showing its typical zinc finger fold. Therefore, our protocol proved adequate for providing correctly folded zinc finger domains.

Despite zinc finger proteins being known to engage in nucleic acid binding, CCHC zinc fingers were shown to favor protein–protein interaction over DNA binding [41]. Members of the FOG protein family are known to interact with the transcription factor family GATA, mediated through the GATA N-terminal zinc finger domain [41]. Dimerization of the transcription factors was previously described. For instance, in Ikaros transcription factors, two C-terminal zinc finger domains participate in dimerization but do not bind DNA [42,43]. Studies showed that BCL11B is also capable of homodimerization [18]. We developed a FRET assay as a useful tool to confirm this dimerization in vitro. The FRET measurements indicated such subunit exchange, verifying the hypothesis of multimerization. The subsequent size exclusion chromatography resulted in elution volumes of 1.35 mL for fluorescently tagged BCL11B_42-94_, which corresponded to complexes 150 kDa in size. This was above the expected dimer size of ~75 kDa and also suggested multimerization. Complexes of tetramer sizes were also found for untagged BCL11B_42-94_. The formation of tetramers of transcription factors was previously shown studying zinc finger protein ZNF350. ZNF350 acts as a transcriptional repressor, which is mediated by BRCA1, and the C-terminal CTRD domain proved to be necessary and sufficient for tetramerization [44].

In order to investigate the oligomerization state of BCL11B_42-94_, protein crystallization and site-directed mutagenesis studies of the domain can prove as helpful techniques. Insights into the PPI interface might reveal residues crucial for the oligomerization of BCL11B_42-94_ and proper function of the protein. Additionally, the crystal structure could verify the proposed ββα fold. Fluorescently-tagged zinc fingers can also be used for DNA microarrays without requiring further labeling steps. The rapid identification of DNA-binding sequences can be achieved. Our optimized protocol achieved up to 100-fold higher yields compared to similar procedures in *E. coli* [45] and will therefore enable such experiments in the future.

## 4. Materials and Methods

### 4.1. Materials

Unless otherwise stated, all chemicals were purchased from Sigma (Sigma-Aldrich, Taufkirchen, Germany). FPLC-associated systems, ÄKTApure, and employed columns, were purchased from Cytiva (Freiburg, Germany). The ÄKTAmicro system was purchased from GE Healthcare (Freiburg, Germany).

### 4.2. Expression Vector Design and Transformation

The codon-optimized N-terminal zinc finger domain of BCL11B (amino acids 42–94) was used to design a fluorescent FRET pair. Therefore, the zinc finger domain was incorporated into a bacterial protein expression vector (pET) with an ampicillin resistance. The whole plasmid, including the BCL11B_42-94_ insert, was designed in and cloned by vectorbuilder (Vectorbuiler Inc., Santa Clara, CA, USA). An N-terminal hexahistidine (His_6_) tag, a cyan fluorescent protein (CyPet) and a yellow fluorescent protein (EYFP) tag, respectively, and a flexible 3x tandem GGGGS linker were introduced. A Tobacco Etch Virus (TEV) cleaving site allowed to obtain the untagged zinc finger domain. KpnI and XhoI restriction sites enabled the insertion of sequences with matching sites. Sequences and successful transformations were verified by sequencing (Microsynth Seqlab, Göttingen, Germany).

The delivered *E. coli* cells, containing the final plasmid, were used for the overnight cultures. DNA was extracted following a standard procedure (GeneJET Plasmid Miniprep Kit, Thermo Fisher, Darmstadt, Germany). The transformation into NiCo21 (DE3) *E. coli* cells was carried out according to the manufacturer’s instructions (New England Biolabs, Frankfurt am Main, Germany). Cells were then plated on lysogeny broth (LB) agar with 100 µg/mL ampicillin. Colonies were excited by UV radiation, and fluorescent colonies were picked after overnight incubation at 37 °C and used for 5 mL overnight cultures (LB with 100 µg/mL ampicillin, 30 °C) shaking. A small amount (0.5 mL) of overnight culture was mixed with 0.5 mL 50% glycerol to prepare glycerol stocks stored at −80 °C. 

### 4.3. Fluorescent Protein Tag Mutagenesis 

An A206K mutation preventing dimerization of the fluorescent tag was introduced by site-directed mutagenesis by QuikChangeXL (Agilent Technologies, Santa Clara, CA, USA). CyPet naturally possessed a lysine at position 206 and could be used without further modification.

### 4.4. Expression of the Recombinant Zinc Finger Domain

Overnight cultures were prepared from glycerol stocks. For each fluorescent construct, two flasks of 0.5 L terrific broth (TB) medium containing 100 µg/mL ampicillin were inoculated 1:25 and grown at 110 rpm and 37 °C. At an OD_600_ = 2, the cultures were induced with 1-mM isopropyl-β-d-thiogalactopyranosid (IPTG). After induction, the expression was carried out for 18 h at 110 rpm and 16 °C. Bacteria were harvested by centrifugation at 4500× *g* and 4 °C for 30 min.

### 4.5. Purification of Fluorescent BCL11B Domain

Cell pellets from a 1 L culture (15 g wet weight) were resuspended in 150 mL buffer containing 20 mM HEPES, 150 mM NaCl, and 1 mM dithiothreitol (DTT), pH 7.4. Lysis was carried out by pulsed sonication (4 cycles, 2 min total time, 0.5 s on/off, and 40% amplitude) using a Branson Digital Sonifier SFX 250 (Emerson, Dietzenbach, Germany). The lysate was clarified by centrifugation and subsequent filtration through 0.22 µM filter membranes (GVS, Sanford, ME, USA).

Consecutive purification steps were carried out on an ÄKTApure platform. The clarified lysate was loaded onto 4 connected HisTrap excel 5 mL columns with 2 mL/min. Theoretically, a bed volume of 20 mL results in a binding capacity of approximately 200 mg His-tagged protein. Due to the oligomerization of BCL11B_42-94_, the binding capacity was increased, as not all the His-tags needed to bind. The columns were washed with 10 column volumes (CV) of equilibration buffer (20 mM HEPES, 150 mM NaCl, and 1 mM DTT, pH 7.4) and 5% elution buffer for 15 CV (20 mM HEPES, 150 mM NaCl, 250 mM imidazole, and 1 mM DTT, pH 7.4) at 5 mL/min. The His-tagged protein was eluted using 100% elution buffer at 2 mL/min. The fractions were collected and analyzed by denaturing SDS-PAGE.

### 4.6. Purification of the Untagged Zinc Finger Domain

The purified fluorescent protein was dialyzed against an equilibration buffer and subsequently digested with TEV protease in a ratio 1:100 (*w*/*w* TEV:protein) at room temperature overnight. The cleavage reaction was loaded onto 2 connected HisTrap excel 5 mL columns at 2 mL/min. The fluorescent tag bound to the column through the His tag, whereas the BCL11B_42-94_ domain remained in the flow-through. After loading, the column was washed with equilibration buffer. The tag was eluted with 100% elution buffer. The fractions were used for analysis by tricine SDS-PAGE for small proteins [46].

### 4.7. Size Determination

The size of fluorescent CyPet-BCL11B_42-94_ was determined using a Superdex 200 Increase 3.2/300 size exclusion chromatography column installed on an ÄKTAmicro platform (Cytiva, Freiburg, Germany). The sizes were determined using a calibration standard curve with the protein standards recommended by the manufacturer.

### 4.8. Structure Determination by Circular Dichroism

Two hundred and fifty micrograms per milliliter of protein in 10 mM Tris, 10 mM NaF, and 1 mM TCEP, pH 7.4 were measured in a 1 mm path-length quartz cuvette (Hellma, Müllheim, Germany) on a Chirascan V100 Circular Dichroism Spectrometer (Applied Photophysics Ltd., Leatherhead, UK). The pH was adjusted using sulfuric acid. CD spectra were recorded from 190 to 250 nm at room temperature. Three repeats for each sample were recorded and averaged. Deconvolution of the raw data was carried out using BeStSel.com (accessed on 18 July 2021) [25,26]. For the CD spectra of BCL11B_42-94_ without zinc, Zn^2+^ was removed by the dialysis of BCL11B_42-94_ against a buffer containing 10 mM glycine-HCl and 1 mM TCEP, pH 2.5 before dialyzing against the CD buffer mentioned above.

### 4.9. Functional FRET Assay

CyPet-BCL11B_42-94_ and mEYFP-BCL11B_42-94_ were mixed 1:1 in the buffer (20 mM HEPES, 150 mM NaCl, and 1 mM DTT, pH 7.4), resulting in a total protein concentration of 200 µg/mL. The solution was incubated at 4 °C for 48 h to ensure complex formation. Fluorescence emission spectra of the mixture and pure CyPet-/mEYFP-BCL11B_42-94_ were measured from 420 to 700 nm at room temperature on a CYTATION 5 plate reader (Biotek, VT, USA). CyPet-BCL11B_42-94_ was excited at 400 nm in opaque 96-well microplates (BRAND GmbH & Co. KG, Wertheim, Germany).

## 5. Conclusions

We developed an easy protocol for the expression and purification of the N-terminal BCL11B domain as a model system for zinc finger domains. It was possible to prepare a functional FRET pair of this domain to examine the oligomerization of BCL11B_42-94_ in vitro. The yield for both fluorescent constructs was very high in comparison to similar purifications procedures carried out in *E. coli* [45]. Due to the integrated restriction sites, this protocol can be utilized to insert any given zinc finger domain sequence for expression and can be used for (i) verifying the putative homodimerization of other zinc fingers, (ii) examining protein–protein-interactions of zinc finger domains with their binding partner, (iii) the identification of DNA sequences that are specifically bound by zinc finger domains using DNA microarrays, and (iv) structural studies requiring high amounts of proteins.

## Figures and Tables

**Figure 1 molecules-26-07576-f001:**
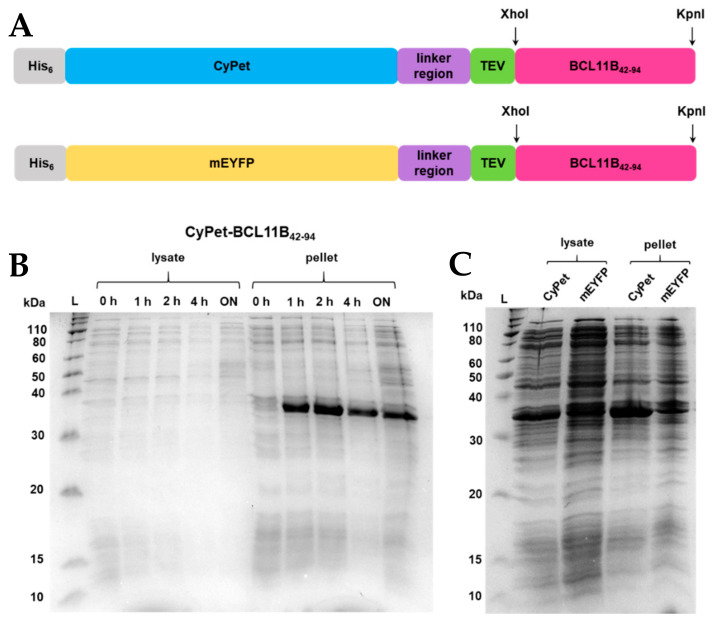
(**A**) BCL11B_42-94_ fusion protein sequence showing an N-terminal His_6_-tag, fluorescent CyPet, and mEYFP protein tag, respectively, a flexible linker region, a TEV-cleaving site, enzyme restriction sites XhoI and KpnI, and the BCL11B_42-94_ zinc finger domain. The BCL11B_42-94_ domain has a size of 5.5 kDa, and the full-length fluorescent fusion proteins have a size of 36 kDa each. (**B**) SDS-PAGE showing expression trials of CyPet- and mEYFP-BCL11B_42-94_ in *E. coli* NiCo21 grown in TB medium at 37 °C at different time points or (**C**) at 16 °C overnight. For comparative reasons, all *E. coli* cultures were diluted to an OD_600_ = 0.6 before lysis and loading the gel.

**Figure 2 molecules-26-07576-f002:**
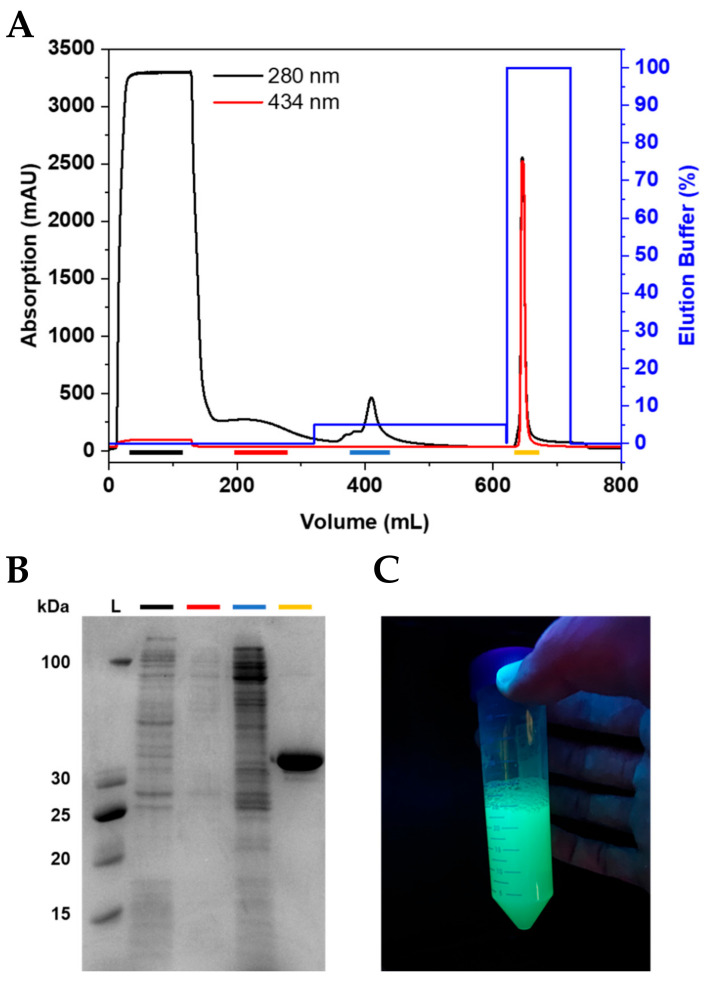
Purification of CyPet-BCL11B_42-94_. (**A**) Chromatogram of CyPet-BCL11B_42-94_ purification using IMAC. (**B**) Corresponding SDS-PAGE showing the collected fractions. The flow-through (black) shows *E. coli* proteins that do not bind to the column. The wash (red) contains loosely bound proteins. The wash step with 5% elution buffer (blue) is crucial to obtain the highly pure fluorescently tagged zinc finger domain (yellow). (**C**) Purified CyPet-BCL11B_42-94_ under UV light.

**Figure 3 molecules-26-07576-f003:**
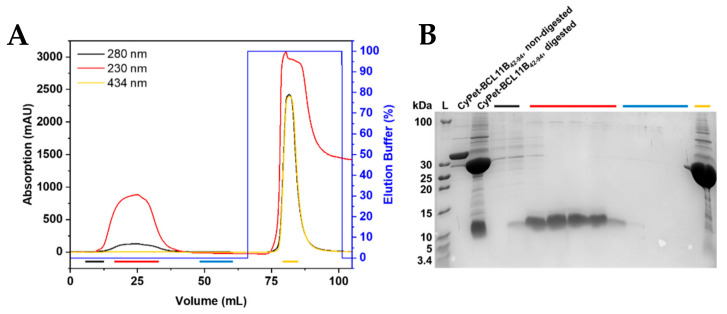
Purification of untagged BCL11B_42-94_. (**A**) IMAC chromatogram of the TEV-digested CyPet-BCL11B_42-94_. The fluorescent tag binds to the column through the His_6_-tag, and the untagged BCL11B_42-94_ elutes. Absorption at 230 nm is used to detect BCL11B_42-94_, whereas absorption at 434 nm corresponds to the CyPet tag. (**B**) Corresponding tricine SDS-PAGE of the collected fractions. Black fractions show the first fractions of the flow-through, with the beginning elution of BCL11B_42-94_ and remaining impurities. Red fractions contain the BCL11B_42-94_ zinc finger domain in high purity and concentration. The wash is shown in blue, with no visible protein content. The CyPet tag elutes with 100% elution buffer and is present in the yellow fraction. Non-digested CyPet-BCL11B_42-94_ was used as a reference at 36 kDa and digested CyPet-BCL11B_42-94_ before purification as a reference for the tag and untagged BCL11B_42-94_.

**Figure 4 molecules-26-07576-f004:**
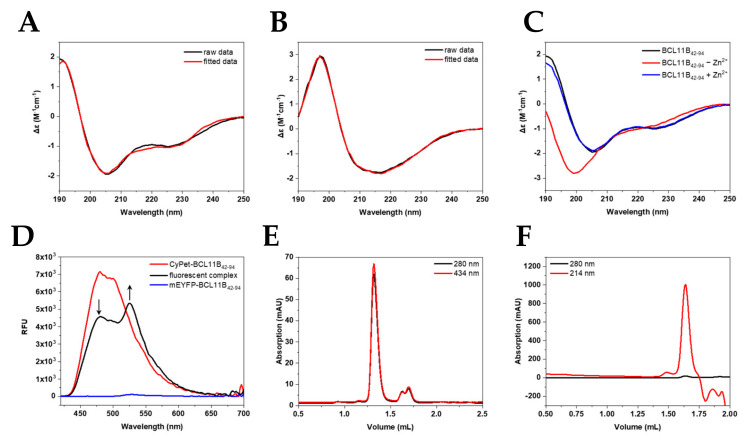
Characterization of the fluorescent BCL11B_42-94_ zinc finger domain. Secondary structure determination of (**A**) untagged BCL11B_42-94_ (**B**) and CyPet-BCL11B_42-94_ using circular dichroism showing raw (black) and fitted data (red). (**C**) CD spectra of BCL11B_42-94_ before (black) and after (red) removing zinc. After adding Zn^2+^ again, the spectrum resembles one of a typically folded zinc finger again. (**D**) FRET of the CyPet-BCL11B_42-94_ and mEYFP-BCL11B_42-94_ oligomers. Donor quenching is indicated by an arrow pointing downwards, while increasing the fluorescence of mEYFP-BCL11B_42-94_ is marked by an arrow pointing upward. (**E**) Size exclusion chromatography of CyPet-BCL11B_42-94_ with an elution volume of CyPet-BCL11B_42-94_ at 1.35 mL corresponding to a larger oligomer formation. (**F**) SEC of untagged BCL11B_42-94_. The elution at 1.64 mL corresponds to complexes around the tetramer size.

**Table 1 molecules-26-07576-t001:** Secondary structure content of BCL11B_42-94_ and its fluorescent constructs. Deconvolution was carried out using BeStSel.com (accessed on 18 July 2021) [25,26]. The secondary structure content is shown in percentage (%). Deconvolution of BCL11B_42-94_ after the TEV digest of mEYFP-BCL11B_42-94_ is shown in Appendix A.

Secondary Structure	BCL11B_42-94_	CyPet-BCL11B_42-94_	mEYFP-BCL11B_42-94_
α-Helix	8.9	12.3	15.7
Antiparallel β-sheet	30.1	25.2	23.3
Parallel β-sheet	0.0	8.1	1.2
β-Turn	15.9	12.3	13.6
Others	45.1	42.1	46.2

## Data Availability

The data presented in this study are available in the present article.

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
