# Peer review of "Easy Expression and Purification of Fluorescent N-Terminal BCL11B CCHC Zinc Finger Domain"

_molecules, 2021, doi:10.3390/molecules26247576_

Round 1

Reviewer 1 Report

Susemihl et al. present a useful lab scale study for the production of a specific zinc finger domain. The proposed protocol is indeed very simple and yields very pure fluorescence-tagged zinc finger domain. The yield is stated to be high by the authors, but no comparison to literature is given. Overall the manuscript "High-Yield Expression and One-Step Purification of Fluorescent BCL11B Zinc Finger Domain" provides a useful methodology and should be published after some flaws are addressed. Detailed line for line comments below.

Title: High-yield is misleading in this case. 200 mg/L is certainly good, but much higher yields have been reported in E. coli (up to 20 g/L). Since the experiments were performed in shaker cultures, 200 mg/L is indeed high, but compared to high cell density fermentation it has to be considered low to mediocre. The words high-yield in the title should be removed or replaced. One-step purification refers to the single chromatography step used for capture, but the manuscript also shows further purification steps with tag removal and additional chromatography. The title might again be somewhat misleading in this regard. Furthermore, only the N-terminal CCHC zinc finger domain of the BCL11B is expressed, making the title even more misleading.

Abstract, line 17: "First, we expressed fluorescent BCL11B42-94 in E. coli and..." This statement is misleading, since BCL11B42-94 is not a fluorescent protein. Please change to fluorescently tagged BCL11B42-94. Continuing in the same sentence, the statement about the purification with subsequent yield statement can be misinterpreted. My understanding is that the 200 mg/L refer to fluorescently tagged protein yield from cell culture, but not yield after purification. Could the authors clarify this?

ln 20: "Validation of the functional FRET pair was achieved 20
with Förster resonance energy transfer." FRET means Förster resonance energy transfer, so this sentence is a tautology and should be reworded.

Introduction, line 20: I have not seen the C2H2 nomenclature before. I have it seen written either Cys2His2 (with subscript) or C2H2 (without subscript). I would think this is done to avoid confusion with acetylene. Please use consistent and clear abbreviations.

ln 42: suggest change "proved to be " to "appears to be". Statement is absolute and also change in tense.

Results section in general: some of the statements should be moved to the Materials and Methods section. The Results section should deal with results only and not describe the experimental procedures in detail. 

ln60: At least the final yields of the CyPet construct should be shown in both sections 2.1 and 2.2 instead of the blanket statement in line 60-61. A lot of the text refers to both constructs already, so the statement seems superfluous. A final statement at the end of 2.1 and 2.2 or only at the end of 2.2 could state that the expression and purification of CyPet and EYFP constructs were essentially the same, but strengthen the statement by stating final expression and purification yields.

ln65: The sequences (amino acid sequence, optionally DNA sequence) should be added as supplementary information. What was the sequence used for the TEV cleaving site? Does TEV cleavage result in an overhang of residual amino acids? If yes, this should be stated somewhere as well.

End of section 2.1: What is the final upstream yield for the protein of interest? Can the authors estimate the amount from the SDS-PAGE? If no yield is known, the statement of high-yield expression in the title becomes guess work.

Figure 1 (D): the total protein concentration of the lysate appears to be very low, especially after 4 hours. Is there an explanation for this?

ln108: Why was the lysate purified on a total of four connected columns? A bed volume of 20 mL should give a capacity of ~200 mg his-tagged protein. I cannot find the exact amount of lysate loaded, but from the chromatogram it appears to be below 200 mL, with a later stated yield of 200 mg/L. This is well below the capacity of the columns. Is this due to low capacity of the column or is the lysate volume of ~150 mL from 1 L fermentation? Material and Methods states 0.5 L. Where all the cells used for the purification? This is not clearly stated in the results section. I would urge the authors to add a table containing the mass balance of both up- and downstream processes, including the tag removal step. This table should contain volumes, concentrations and yields.

ln124: while the possibility to cleave the tag is mentioned, a statement on the use of TEV protease should be added to the sentence about removal of the tag. The cleavage is currently only implied (and of course described in material and methods).

ln132: tricine PAGE should be tricine SDS-PAGE or SDS-PAGE

ln133: why is the purity of the initial breakthrough (black) lower than the main portion of the flow-through fraction (red). Why did 6H-tagged CyPet bind to the column if the elution fraction was loaded. The elution fraction contains high concentrations of imidazole and should therefore not adsorb in its entirety.

ln175: I find the interpretation of the SEC data lacking. Can the authors add data on the standard material that was injected to infer the 150 kDa size of the oligomer into the supplementary materials? What about the minor peaks around 1.6 - 1.7 mL? Are those dimer and monomer or product fragments? The SDS-PAGE in 2B shows minor bands directly underneath the main product. Can the authors be sure that no product related impurities in the form of fragments are present? Do these fragments pose a problem for the proposed purpose of this material? Can these impurities be observed in either N-terminal or C-terminal part of the cleavage product?

Figure 4: The results of EYFP-BCL should be shown for comparison or at least added to the supplementary materials. Otherwise the initial statement in ln60 cannot be confirmed.

ln204: How do the authors explain the similarity in CD results between BCL and EYFP-BCL but not CyPet-BCL? Data on the structure of the CyPet and EYFP cleaved constructs would be helpful for comparison here, otherwise the data does not directly support the conclusion in line 207.

ln220: the observed tetrameric structure of the fluorescence tagged BCL appears to be unexpected. Can the authors provide data on the multimerization state of BCL after tag removal?

ln324: no references are cited to justify this statement. What yields are to be expected from similar purifications in microbial systems? This should be added to the discussion.

Reviewer 2 Report

The manuscript by Anne Susemihl et al. describes a method for High-Yield Expression and One-Step Purification of fluorescent protein labelled BCL11B Zinc Finger (ZF) Domain. It follows the work referenced at number 20 work on the BCL11B by one of the co-authors. The manuscript reads well and provides method for high yield production of small domains, however some improvements are needed. It has been assigned to Special Issue on Separation and Purification of Peptides, therefore I would welcome additional information in this regard in introduction, results and discussion. As a scientist in the field of metal proteins I would also like to see more zinc-related data. Just a small addition so it would not divert from the main focused proposed above. I particularly like the color coding of chromatogram fractions to gel lanes, which allows to quickly compare results.

In details:

-  The target N-terminal BCL11B domain is 52 amino acid long. This is out of the comfort length for chemical peptide synthesis but not impossible. Why do you propose the use of bacterial expression system? Please add few sentences.

- There is a number of proteins that can be used as a fusion for expression e.g. MBP, GST. Why is GFP derivatives preferred? You state that the yield is 0.2g/L is it high/low compared to other papers describing the expression of peptides/domains with different fusions? Please add few sentences.

- Why there is a DTT in the lysis/column buffer? The DTT can strip the Ni ions from the column forming a brown color complex, therefore it is not recommended by the manufacturers. The purification works, as we see on Fig. 2B however the yield could be drastically affected. Please comment.

- Figure 3 – I wonder about the delayed elution from HisTrap column of the BCL11B domain after TEV cleavage. Does the column has a void volume of ~10 ml, or actually the zinc finger domain through its His and Cys residues is able to form a weak interaction with immobilized Ni on the column, therefore slowing down the elution? On the same figure panel B, I find it interesting that the domain released after TEV cutting is ~12 kDa in size, whereas we expect 5.5 kDa. Could you comment on that? Since it is a denaturing gel I do not expect any dimers.

- I would welcome some test to prove that the domain is functional i.e. it binds zinc ions. On the purified domain you could do a test on CD by adding zinc ions followed by EDTA to observe the changes in the secondary structures. You could do the same test (adding zinc, followed by EDTA) on the FRET assay to see whether zinc ion is necessary for the homodimer formation of the BCL11B (in part this was proven in ref 20 in cell based studies). In the FRET assay it seems that since no zinc ions are added they are not necessary for the process. However it is also possible that the proteins expressed in bacteria already have the zinc ion bound. In my experience it is possible when zinc ions forms highly stable complex with the protein/domain (pKd > 13). When making this experiments please avoid DTT, as it will compete for zinc ion with the domains. As a reducing agent I recommend to use TCEP.

Minor issues:

Figure 1 – I suggest moving panels B and C to supporting information.

Page 2 lines 51-56 – The fragment on broad use of ZF seems out of place here as below and above the text describes the BCL11B specifically. I suggest moving this part to discussion to show why it is needed to have optimized solutions for purification of ZFs.

Page 2 line 56 – the double use of “and” in the sentence. Use comma or modify the sentence.

Page 7 line 238 – the end of the sentence on GE aqusition by Cytiva is not necessary

Page 7 line 242 – how was the gene of ZF domain of BCL11B obtained? Was it synthesized or obtained via PCR from other plasmid? Please provide the details.

Page 8 line 259 – the sentence about sequencing should be moved to line 250 as this is the end of plasmid description.

Page 8 line 290 – please describe the buffer composition for TEV protease cleavage or indicate the manufacturer's buffer if it is a commercial product.

Page 9 – the placement of the conclusions after methods seems odd. I would put them directly after discussion.

Round 2

Reviewer 1 Report

All issues were addressed and the paper should be accepted in the present form.